# *Acinetobacter baylyi* regulates type IV pilus synthesis by employing two extension motors and a motor protein inhibitor

Courtney K. Ellison [1,2✉], Triana N. Dalia [3], Catherine A. Klancher[3], Joshua W. Shaevitz [1✉], Zemer Gitai [2✉] & Ankur B. Dalia [3✉]

Bacteria use extracellular appendages called type IV pili (T4P) for diverse behaviors including DNA uptake, surface sensing, virulence, protein secretion, and twitching motility. Dynamic extension and retraction of T4P is essential for their function, and T4P extension is thought to occur through the action of a single, highly conserved motor, PilB. Here, we develop *Acinetobacter baylyi* as a model to study T4P by employing a recently developed pilus labeling method. By contrast to previous studies of other bacterial species, we find that T4P synthesis in *A. baylyi* is dependent not only on PilB but also on an additional, phylogenetically distinct motor, TfpB. Furthermore, we identify a protein (CpiA) that inhibits T4P extension by specifically binding and inhibiting PilB but not TfpB. These results expand our understanding of T4P regulation and highlight how inhibitors might be exploited to disrupt T4P synthesis.

[1] Lewis-Sigler Institute for Integrative Genomics, Princeton University, Princeton, NJ, USA. [2] Department of Molecular Biology, Princeton University, Princeton, NJ, USA. [3] Department of Biology, Indiana University, Bloomington, IN, USA. ✉email: c.ellison@princeton.edu; shaevitz@princeton.edu; zgitai@princeton.edu; ankdalia@indiana.edu

T4P are thin, proteinaceous appendages that are broadly distributed throughout bacteria and archaea[1,2]. T4P are composed primarily of major pilin protein subunits that are polymerized or depolymerized through the activity of ATPases to mediate fiber extension and retraction, respectively[3]. T4P are subdivided into subcategories (generally T4aP, T4bP, and T4cP) based on protein homology and pilus function, with T4aP being the best characterized. In the T4aP systems found in Gram-negative organisms, polymerization and depolymerization of pilins occurs through interactions of the extension ATPase PilB or retraction ATPase PilT with the integral inner membrane platform protein PilC (Supplementary Fig. 1a). The growing fiber spans an alignment complex from the inner membrane through the periplasm composed of PilNOP to exit through the PilQ outer membrane secretin pore. Dynamic cycles of T4P extension and retraction are critical for the diverse processes that these structures mediate including twitching motility[4], surface sensing[5,6], virulence[7,8], and DNA uptake[9,10].

*Acinetobacter* species like *A. baumannii* and *A. nosocomialis* have emerged to become an urgent medical threat due to their prevalence in hospital-acquired infections and their capacity to acquire antibiotic resistance (AbR) genes; a process that is achieved in part by natural transformation through T4aP-mediated DNA uptake[11,12]. *Acinetobacter baylyi* is the most naturally transformable species reported to date[13], with up to 50% of cells undergoing natural transformation in laboratory conditions, making it an ideal candidate to study T4P-mediated DNA uptake and natural transformation.

Here, we develop *A. baylyi* as a new model for studying T4P dynamics. We show that *A. baylyi* uses two phylogenetically distinct motor proteins to drive T4P extension and a motor protein inhibitor to control T4P synthesis. These findings may be relevant for other T4P and highlight the value of developing new

model systems for the study of these broadly conserved surface appendages.

## Results

**Tn-seq reveals factors important for natural transformation and T4P synthesis.** To study *A. baylyi* T4P, we applied a recently developed labeling method[5,14] by targeting the major pilin, ComP, for cysteine substitution and subsequent labeling with thiol-reactive maleimide dyes. Maleimide-labeling of the functional *comP*[T129C] strain (Supplementary Fig. 1b, c and Supplementary Table 1) revealed external T4P filaments as seen in other species using this method[14]. T4P in *A. baylyi* are much shorter, often appearing as puncta rather than the 1-μm long filaments found in other species like *V. cholerae*[10] or the 10-μm long filaments found in *Legionella pneumophila*[15]. *A. baylyi* TFP also localize close together in a line along the long axis of the cell, although the functional consequences of this localization pattern remain unclear (Supplementary Fig. 1c). Recent work has shown that T4P synthesis in *A. baumannii* is growth-phase dependent[16], but we find here that the T4P of *A. baylyi* are constitutively made (Supplementary Fig. 2), in line with previous findings that *A. baylyi* cells are transformable throughout all growth phases[17].

Incubation of cells with fluorescently labeled DNA resulted in co-localization of DNA with T4P, which is consistent with the essential role of T4P-DNA binding during natural transformation in *Vibrio cholerae*[10] (Fig. 1a). We reasoned that natural transformation could be used to screen for other factors that regulate T4P synthesis in *A. baylyi*. To that end, we performed a high-throughput transposon-sequencing screen (Tn-seq)[18] to identify genes required for natural transformation as done previously in other species[15,19,20]. In this screen, loss of known T4P-related genes resulted in negative selection, indicating that

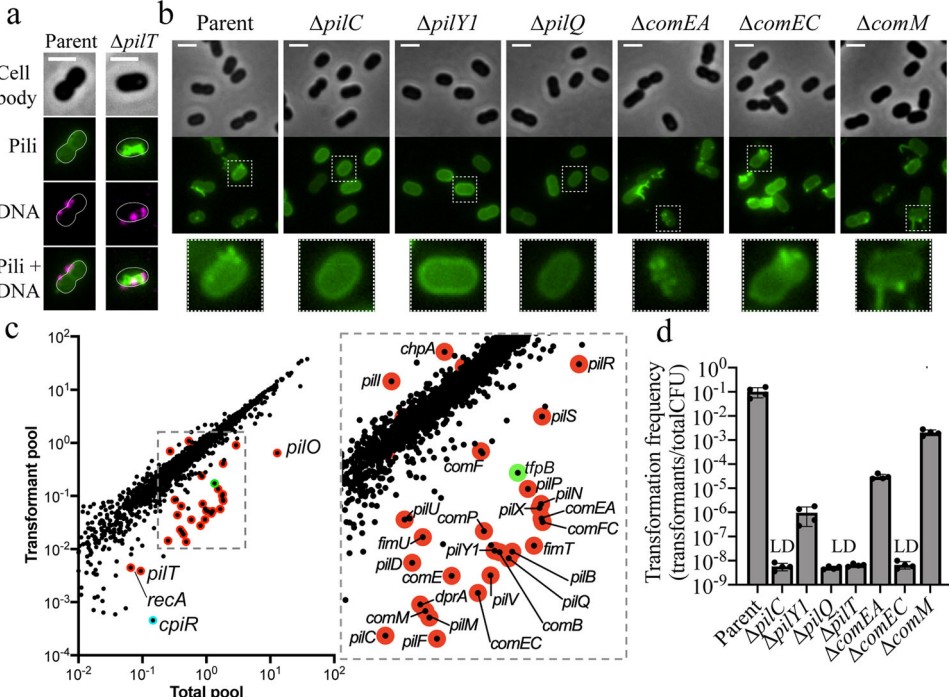

**Fig. 1 Tn-seq reveals factors important for natural transformation and T4P synthesis. a** Representative images of cells with AF488-mal-labeled pili incubated with fluorescently labeled DNA. **b** Representative images of *pilT*-intact strains from natural transformation assays shown in **d** labeled with AF488-mal. Zoomed-in images of representative single cells from each strain are outlined in dashed boxes and shown below. Scale bars, 2 μm. **c** Visual representation of Tn-seq screen showing the relative abundance of each gene in the "total pool" of transposon mutants compared to the "transformant pool" recovered after natural transformation. Known T4P structural/regulatory genes are outlined in red. *tfpB* is outlined in green, and *cpiR* is outlined in cyan. **d** Natural transformation assays of indicated strains. Each data point represents a biological replicate (*n* = 4) and bar graphs indicate the mean ± SD. LD limit of detection. Source data are provided as a Source Data file.

they were critical for natural transformation as expected (Fig. 1c, Supplementary Data 1, and Supplementary Table 1). To validate our Tn-seq results, we made in-frame deletions of representative genes from each T4P-encoding operon, as well as genes known to be essential for natural transformation that act downstream of T4P. Mutations in the T4P platform protein gene *pilC*, the pilus regulatory gene *pilY1*, the outer membrane secretin gene *pilQ*, and the retraction ATPase gene *pilT* all showed a marked reduction in natural transformation, with most mutants exhibiting transformation rates below our limit of detection (Fig. 1d). T4P labeling of *pilC*, *pilY1*, and *pilQ* mutants revealed no visible T4P fibers (Fig. 1b). Mutations in genes that act downstream of T4P, including the periplasmic DNA-binding protein gene *comEA*, the inner membrane DNA-transporter protein gene *comEC*, and the DNA-recombination helicase gene *comM* likewise resulted in a reduction in transformation frequency; however, these strains still produced T4P fibers as expected (Fig. 1b, d).

**TfpB is a distinct PilB homolog required for efficient T4P extension.** We next sought to determine whether any of the previously uncharacterized genes from our Tn-seq screen reduced natural transformation in *A. baylyi* by affecting T4P synthesis. Tn-seq revealed that mutations in *pilB*, the canonical extension ATPase gene that is typically co-transcribed with *pilC* and the pre-pilin peptidase, *pilD*, resulted in reduced natural transformation (Fig. 1c). We also found an additional *pilB* homolog that likewise exhibited lower rates of transformation (Fig. 1c). We named this new PilB homolog TfpB for "type four pilus PilB-like protein" because although it has homology to PilB, it does not exhibit the same gene synteny as *pilB*, which is typically co-transcribed with *pilC*, the gene encoding the cognate inner membrane platform protein (Supplementary Fig. 3). Surprisingly, deletion of the canonical *pilB* did not ablate transformation as would be expected based on homology to other T4P[9] (Fig. 2a). Mutation of *tfpB* reduced transformation rates by two orders of magnitude, similar to the *pilB* mutant; however, natural

transformation was undetectable in the *pilB tfpB* double mutant (Fig. 2a).

T4P labeling in *pilB*, *tfpB*, and *pilB tfpB* mutants revealed that none of these three mutants produced visible pili while *pilT* was intact (Fig. 2c). Because T4P are required for natural transformation, we hypothesized that *pilB* or *tfpB* mutants, which are still highly transformable, may either (1) make short pili that cannot be resolved by our fluorescence microscopy-based approach or (2) exhibit reduced rates of pilus dynamic activity that result in a reduction in detectable surface pili in static images. To test the latter hypothesis, deletion of *pilT* prevents T4P retraction[21] and can thus reveal more subtle effects on pilus extension since all T4P made by a cell remain in an assembled state. In a *pilT* deletion background, *pilB* mutant populations had similar numbers of cells producing T4P compared to the Δ*pilT* parent while, surprisingly, *tfpB* mutants were highly defective in T4P synthesis (Fig. 2b, c). The *pilB tfpB pilT* triple mutant produced no detectable T4P fibers (Fig. 2b, c). These data suggest that PilB and TfpB may play distinct roles in T4P extension. Because *tfpB pilT* mutants make very few T4P, but *pilB pilT* mutants make similar levels of surface pili compared to the Δ*pilT* parent, we hypothesize that TfpB plays a larger role in initiating T4P extension, while PilB plays a larger role in driving processive T4P extension. In this model, each motor must still be able to partially compensate for the loss of the other to achieve some degree of T4P extension, which would account for the observed similarities in natural transformation frequencies when comparing *pilB* and *tfpB* single mutants. While T4P labeling in *pilT* mutant backgrounds supports this model (Fig. 2b, c), it does not rule out the possibility that *tfpB pilT* mutants produce many short T4P that cannot be resolved by our pilus labeling approach. To test this, we incubated extension motor mutants with fluorescently labeled DNA, reasoning that DNA binding would reveal short T4P synthesized in *tfpB pilT* mutants. Fluorescent DNA did not bind the *tfpB pilT* mutant above the background level observed in the *pilB tfpB pilT* mutant except where T4P were visible (Supplementary Fig. 4), suggesting that short pili are not produced in the *tfpB pilT* strain, which further supports the

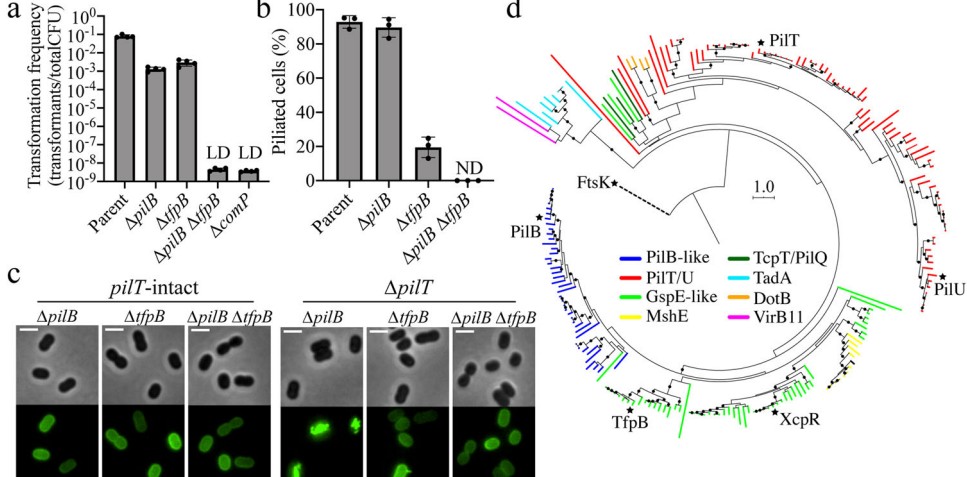

**Fig. 2 TfpB is a phylogenetically distinct PilB homolog that is required for efficient T4P extension in *A. baylyi*. a** Natural transformation assays of indicated strains. Each data point represents a biological replicate (*n* = 4) and bar graphs indicate the mean ± SD. LD limit of detection. ComP is the major pilin in *A. baylyi*, and the mutant is used here as a negative control. **b** Percent of piliated cells in *pilT* mutant populations of indicated strains. Each data point represents an independent, biological replicate (*n* = 3) and bar graphs indicate the mean ± SD. For each biological replicate, a minimum of 70 total cells were assessed. ND no pili detected. **c** Representative images of indicated strains labeled with AF488-mal with background fluorescence subtracted. Scale bars, 2 μm. **d** A rooted phylogeny of TfpB homologues found among Gammaproteobacteria. Branches are colored according to protein annotations in IMG, and nodes with bootstrap values greater than or equal to 70% are indicated by black circles. Black stars are at the tips of branches representing *A. baylyi* proteins with indicated protein names. Source data are provided as a Source Data file.

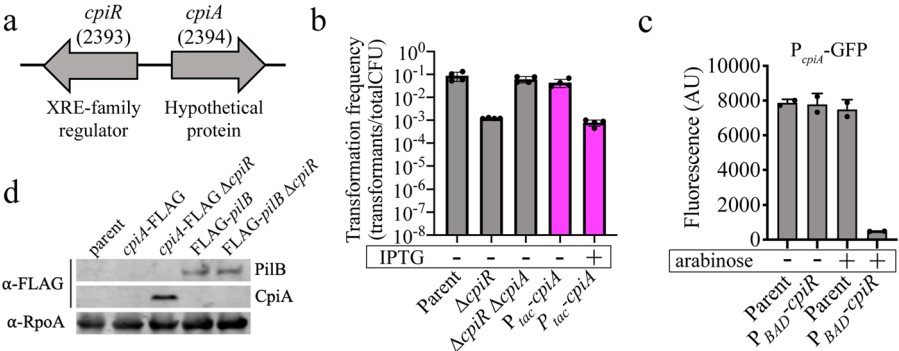

**Fig. 3 Natural transformation is regulated by the inhibitor protein, CpiA, whose expression is controlled by the transcriptional repressor, CpiR.**
**a** Schematic of *cpiRA* locus organization. Numbers in parentheses are ACIAD numbers associated with each gene. **b** Natural transformation assays of the indicated strains performed with or without 100 μM IPTG added as indicated. Each data point represents a biological replicate (*n* = 4) and bar graphs indicate the mean ± SD. Pink bars denote a strain where *cpiA* is expressed at an ectopic location under the control of an IPTG-inducible promoter.
**c** Fluorescence intensity of *V. cholerae* strains harboring a $P_{cpiA}$-GFP reporter and a $P_{BAD}$-*cpiR* ectopic expression construct as indicated when grown with or without 0.2% arabinose as indicated. Each data point represents a biological replicate (*n* = 2) and bar graphs indicate the mean ± SD. **d** Western blot showing CpiA-FLAG production in the indicated strains and PilB levels when CpiA is expressed. RpoA was detected as a loading control. Source data are provided as a Source Data file.

model that TfpB plays a distinct role from PilB in initiating T4P extension.

Alternatively, it is possible that these motors are completely redundant, and the difference in phenotypes observed between *pilB* and *tfpB* mutants is simply due to a dose-dependent loss of motor protein expression. To test this, we sought to determine whether overexpression of *tfpB* was sufficient to compensate for a *pilB* deletion or whether overexpression of *pilB* was sufficient to compensate for a *tfpB* deletion. If these motors are completely redundant, we expected that overexpression of a motor would compensate for loss of the other motor and result in parental levels of natural transformation; however, if these motors play distinct roles in T4P extension, overexpression would not be able to cross compensate and transformation frequencies would remain unchanged. When we performed this experiment, we found that overexpression of *pilB* did not restore the transformation defect of a *tfpB* mutant, and correspondingly, overexpression of *tfpB* did not restore the transformation defect of a *pilB* mutant (Supplementary Fig. 5). Together, these results indicate that both PilB and TfpB are essential for optimal T4P production in *A. baylyi*, with each motor playing a distinct role in T4P extension.

Phylogenetic analysis of extension ATPase homologues found among Gammaproteobacteria[1,22], which include members of the PilT/VirB11 family of secretion ATPases, revealed that TfpB clusters with a group of proteins that are phylogenetically distinct from the canonical PilB ATPase (Fig. 2d and Supplementary Fig. 6). PilB and TfpB proteins are as divergent from one another as PilB and the type II secretion system proteins XcpR/GspE, or PilB and the MshE motors that drive mannose-sensitive haemagglutinin pilus synthesis, suggesting that TfpB evolved as a functionally divergent class of proteins that is distinct from canonical PilB extension motors. TfpB is conserved in the other *Acinetobacter* species analyzed here (Fig. 2d and Supplementary Fig. 6), implying that the use of multiple extension motors may be prevalent in the *Acinetobacter* clade. These results suggest that proteins that cluster with TfpB may play similar functions in other bacterial species and that multiple extension ATPases may be a common feature of diverse T4P.

**Natural transformation is regulated by the inhibitor protein, CpiA.** In addition to revealing the importance of TfpB in T4P synthesis, our Tn-seq results also uncovered an uncharacterized transcriptional regulator belonging to the XRE-family

of transcriptional repressors that is critical for natural transformation (Figs. 1c and 3a). Deletion of this regulator, named *cpiR* for competence pilus inhibition repressor, resulted in a 100-fold reduction in transformation (Fig. 3b). We hypothesized that CpiR repressed a factor that inhibits natural transformation. Transcriptional repressors are often transcribed immediately adjacent to the genes they repress[23,24]. We thus deleted the upstream gene, named *cpiA* for competence pilus inhibition actuator (Fig. 3a), and found that transformation frequency was restored in the *cpiR cpiA* double mutant background (Fig. 3b). Expression of *cpiA* under the control of an IPTG-inducible promoter ($P_{tac}$) was sufficient to reduce transformation frequency even in the presence of *cpiR*, further suggesting that CpiR represses *cpiA* transcription (Fig. 3b). Furthermore, we found that CpiR was sufficient to repress the *cpiA* promoter (using a $P_{cpiA}$-GFP reporter) in the heterologous host *Vibrio cholerae*, which suggests that CpiR is a direct repressor of $P_{cpiA}$ (Fig. 3c). Finally, we found that CpiA protein is only produced in a *cpiR* mutant (Fig. 3d). These results demonstrate that under lab conditions, CpiR represses *cpiA* transcription to allow for high rates of natural transformation; however, in the absence of CpiR, CpiA is produced and natural transformation is inhibited ~100-fold.

CpiA lacks primary sequence or structural homology to any proteins or domains of known function[25,26], so we next sought to determine the mechanism of transformation inhibition by CpiA. Labeling of T4P in a *cpiR* mutant (i.e., when CpiA is expressed) revealed that cells were deficient in T4P synthesis (Fig. 4a). This defect was dependent on *cpiA* because piliation was restored in the *cpiR cpiA* double mutant (Fig. 4a). Deletion of *pilT* restored T4P synthesis in the *cpiR* mutant (i.e., the *cpiR pilT* double mutant) (Fig. 4a, b). This phenotype was reminiscent of the restoration of piliation observed in the *pilB pilT* double mutant, so we hypothesized that CpiA may regulate natural transformation by inhibiting PilB.

**CpiA inhibits PilB activity.** To test if CpiA functions by inhibiting PilB, we made *pilB cpiR* and *tfpB cpiR* double mutants. T4P synthesis was unaffected in *pilB cpiR pilT* (where TfpB is the sole extension ATPase present), while no T4P were detected in *tfpB cpiR pilT* (where PilB is the sole extension ATPase present) (Fig. 4a, b). The defect in T4P synthesis in the *tfpB cpiR pilT* strain was restored when *cpiA* was deleted in this background,

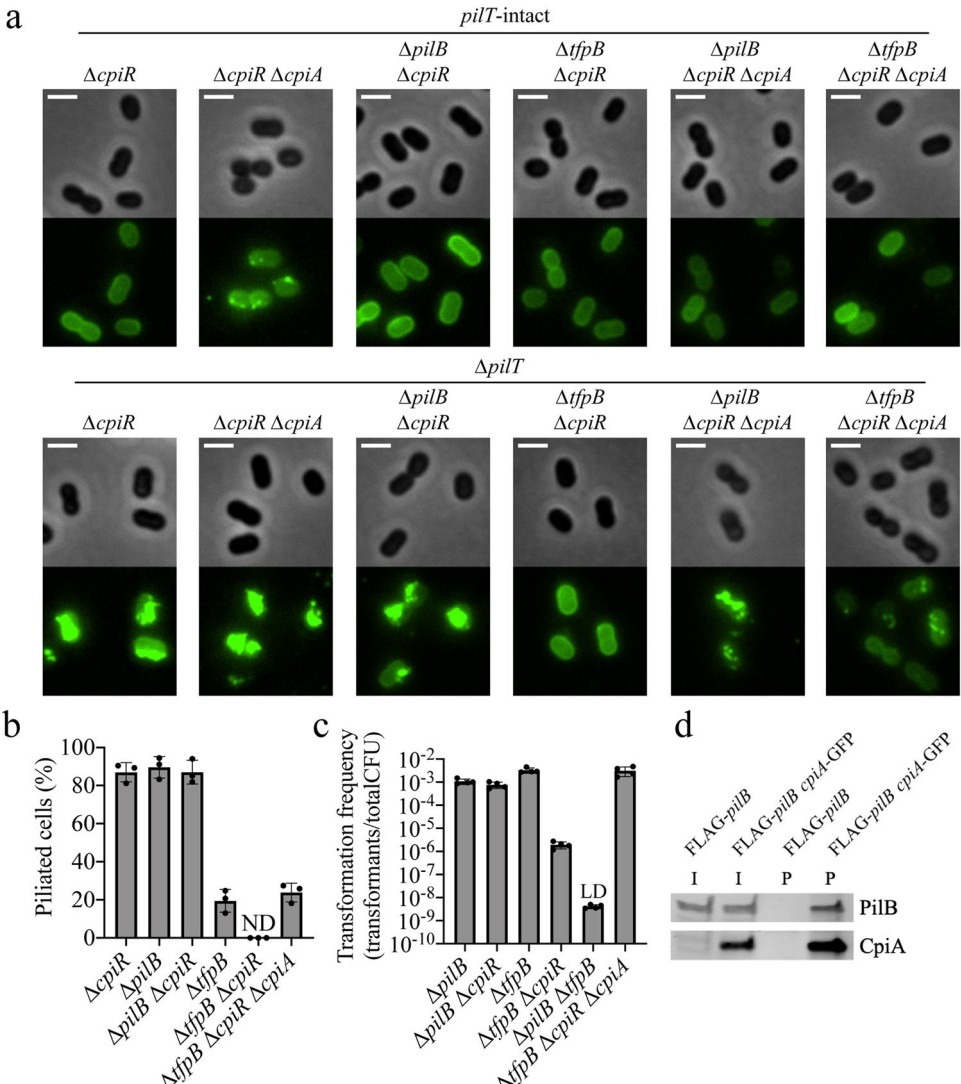

**Fig. 4 CpiA inhibits PilB activity. a** Representative images of indicated strains labeled with AF488-mal with background fluorescence subtracted. Scale bars, 2 μm. **b** Percent of piliated cells in *pilT* mutant populations of indicated strains. Each data point represents an independent, biological replicate ($n = 3$) and bar graphs indicate the mean ± SD. For each biological replicate, a minimum of 70 total cells were assessed. ND no pili detected. **c** Natural transformation assays of indicated strains. Each data point represents a biological replicate ($n = 4$) and bar graphs indicate the mean ± SD. LD limit of detection. **d** Western blot showing coimmunoprecipitation experiments where CpiA-GFP was used as the bait protein to test for interaction with FLAG-PilB as the prey protein in Δ*cpiR* mutant backgrounds. I input sample, P pulldown (coimmunoprecipitation) sample. Source data are provided as a Source Data file.

demonstrating that CpiA acts as a PilB inhibitor to control natural transformation (Fig. 4a, b). Transformation frequency assays corroborated these results by demonstrating that deletion of *cpiR* only inhibited transformation in the *tfpB* mutant background (where PilB is the sole extension ATPase) and not the *pilB* mutant background (where TfpB is the sole extension ATPase) (Fig. 4c). Immunoblotting revealed that CpiA does not affect PilB production or stability (Fig. 3d), and we thus hypothesized that CpiA may interact with PilB to disrupt its function. Coimmunoprecipitation experiments with functional fusion proteins (Supplementary Figs. 7 and 8) revealed that CpiA and PilB interact with each other, and that their interaction is not dependent on the pilus machinery proteins PilMNOPQ (Fig. 4d and Supplementary Fig. 8), ruling out indirect binding of CpiA to PilB through other machinery components required for T4P synthesis. CpiA and TfpB do not interact (Supplementary Fig. 8), consistent with our genetic evidence that CpiA specifically inhibits PilB and not TfpB (Fig. 4a–c). Together, these data suggest that CpiA directly binds to and inhibits PilB.

## Discussion

Environmental conditions play an integral role in regulating how bacteria respond to and interact with their surroundings[27]. The mechanisms of T4P synthesis regulation identified here likely reflect how environmental conditions influence bacterial physiology. The acquisition of the additional extension motor TfpB and the PilB-specific inhibitory protein CpiA likely resulted from a need to modulate T4P extension under different environmental conditions. Bacteria generally limit T4P synthesis to environments where they provide a selective advantage. For example, in *V. cholerae*, the production of competence T4aP requires chitin and quorum sensing[28], which are environmental conditions that cells experience when they are likely to encounter other bacterial cells for horizontal gene transfer, while the toxin-coregulated T4bP required for intestinal colonization are produced in response to host-specific cues[29]. In *A. baylyi*, both PilB and TfpB are required for optimal T4P extension and, consequently, efficient natural transformation. While CpiA is not expressed under laboratory conditions (due to CpiR repression), it is possible that

different environmental conditions derepress *cpiA* to decrease pilus activity.

The discovery that two extension motors drive T4P synthesis in *A. baylyi* raises an interesting biological question. How might two extension motors engage with the machinery during extension? Previous work has shown that multiple retraction motors can work together to drive forceful T4P retraction[21,30], and while the nature of how these motors are able to simultaneously interact with the machinery remains unknown, a similar mechanism may exist for PilB and TfpB. Alternatively, each motor may play temporally distinct roles during extension that allow each motor complex to interact with the machinery separately. Our data show that *tfpB* mutants make substantially fewer pili than *pilB* mutants in the absence of retraction, suggesting that TfpB may play a role in initiating T4P extension while PilB may promote processive pilus extension. In this model, each motor possesses a distinct function that could be accomplished by temporally segregated machinery interactions. Interestingly, our phylogenetic analysis demonstrated that 15 of the 16 species that contained a TfpB motor homolog also encoded a PilB motor homolog (Supplementary Table 2). The exception to this observation, *Pseudomonas fluorescens* A506, possesses T4P machinery components, but lacks a *pilB* homolog and instead only harbors a *tfpB* homolog (Supplementary Fig. 3). This could suggest that TfpB is capable of acting as the sole extension motor in this species, though data are needed to test this hypothesis.

Secretion ATPases like PilB belong to a broadly distributed class of proteins that play essential roles in diverse microbial behaviors, yet there are few known inhibitors of these proteins. The ones that are characterized are encoded by phages that use T4P to infect cells, and it is speculated that PilB inhibition by these proteins prevents superinfection by other T4P-dependent phages[31,32]. CpiA is poorly modeled by structural prediction algorithms like Phyre2[25]. However, it is predicted that CpiA has a secondary structure rich in alpha helices, similar to a recently described phage-encoded PilB inhibitor, Aqs1[32]. Although Aqs1 and CpiA are dissimilar at the sequence level, it is possible that they share structural similarities that enable PilB binding and inhibition. In *Acinetobacter baylyi*, *cpiA* is encoded on the chromosome, and it does not appear to be found in other sequenced species of *Acinetobacter*. Thus, it is tempting to speculate that CpiA was acquired after phage infection of an ancestral strain and co-opted for regulation of T4P activity. While expression of phage-encoded PilB-inhibitors like Aqs1 is not repressed in the same manner as CpiA, it could be that the CpiR repressor was acquired after CpiA to regulate its effects on T4P extension.

The molecular mechanism of CpiA-dependent inhibition of PilB remains unclear. We demonstrate that CpiA binds to PilB. Thus, it is possible that (1) CpiA sequesters PilB to prevent it from interacting with machinery components, (2) that CpiA binds PilB to inhibit its ATPase activity, or (3) that CpiA modifies PilB to disrupt its function. Testing these hypotheses will be the focus of future work. A better understanding of secretion ATPase function and the evolution of mechanisms by which they can be inhibited may enable the development of tools to control their activity. Precise control of T4P synthesis may provide a means to manipulate behaviors that require T4P, like biofilm formation, virulence, and natural transformation, which are clinically relevant in diverse pathogens.

## Methods

**Bacterial strains and culture conditions**. *A. baylyi* strain ADP1 was used throughout this study. For a list of strains used throughout, see Supplementary Table 3. *A. baylyi* cultures were grown at 30 °C in Miller lysogeny broth (LB) medium and on agar supplemented with kanamycin (50 μg/mL), spectinomycin

(60 μg/mL), gentamycin (30 μg/mL), and/or chloramphenicol (30 μg/mL), rifampicin (30 μg/mL), zeocin (100 μg/mL), apramycin (30 μg/mL), and/or streptomycin (10 μg/mL) as appropriate.

**Construction of mutant strains**. Mutants in *A. baylyi* were made using natural transformation as described previously[21]. Briefly, mutant constructs were made by splicing-by-overlap (SOE) PCR to stitch (1) ~3 kb of the homologous region upstream of the gene of interest, (2) the mutation where appropriate (for deletion by allelic replacement with an AbR cassette, or the fusion protein), and (3) ~3 kb of the homologous downstream region. For a list of primers used to generate mutants in this study, see Supplementary Table 4. The upstream region was amplified using F1 + R1 primers, and the downstream region was amplified using F2 + R2 primers. All AbR cassettes were amplified with ABD123 (ATTCCGGGGATCCGTCGAC) and ABD124 (TGTAGGCTGGAGCTGCTTC). Fusion proteins were amplified using the primers indicated in Supplementary Table 4. In-frame deletions were constructed using F1 + R1 primer pairs to amplify the upstream region and F2 + R2 primer pairs to amplify the downstream region with ~20 bp homology to the remaining region of the downstream region built into the R1 primer and ~20 bp homology to the upstream region built into the F2 primer. SOE PCR reactions were performed using a mixture of the upstream and downstream regions, and middle region where appropriate using F1 + R2 primers. SOE PCR products were added with 50 μL of overnight-grown culture to 450 μL of LB in 2-mL round-bottom microcentrifuge tubes (USA Scientific) and grown at 30 °C rotating on a roller drum for 1–3 h. For AbR-constructs, transformants were serially diluted and plated on LB and LB + antibiotic. For in-frame deletions and protein fusion constructs, after the 3-h incubation, cells were incubated with 10 μL of DNase I (New England Biolabs) for 5 min at room temperature before 450 μL of LB was added. Transformations were grown for an additional hour before cells were diluted and 100 μL of 10⁻⁶ dilution was plated on LB plates. In-frame deletions were confirmed by PCR using primers ~150 bp up- and downstream of the introduced mutation, and fusions were confirmed by sequencing. Complementation strains were constructed by placing the gene of interest under a constitutive P$_{tac}$ promoter at the *vanAB* locus, which has previously been established as a site for chromosomal expression[33]. First, the *vanAB* genes were replaced with a *kanR* cassette as described above. Then, a P$_{tac}$ promoter was introduced using primers CE317 + CE260 to amplify the upstream region, CE261 + CE336 to amplify the P$_{tac}$ promoter, CE1205 + CE1206 to amplify *pilB* or CE1207 + CE1208 to amplify *tfpB*, and CE406 + CE176 to amplify the downstream region. SOE PCRs and natural transformation were then performed exactly as described above.

Mutants in *V. cholerae* were made by chitin-induced natural transformation exactly as previously described[34,35]. SOE products were generated exactly as described above. The chromosomally integrated P$_{BAD}$ ectopic expression construct is described in detail in ref. [36].

**Natural transformation assays**. Assays were performed exactly as previously described[37]. Briefly, strains were grown overnight in LB broth at 30 °C rolling. Then, ~10⁸ cells were subcultured into fresh LB medium and 100 ng of tDNA was added. In this study, a ΔACIAD1551::Spec$^R$ or a ΔACIAD1551::Apramycin$^R$ PCR product (with 3 kb arms of homology) was used as the tDNA. Reactions were incubated with end-over-end rotation at 30 °C for 5 h and then plated for quantitative culture on spectinomycin plates (to quantify transformants) and on plain LB plates (to quantify total viable counts). Data are reported as the transformation frequency, which is defined as the (CFU/mL of transformants) / (CFU/mL of total viable counts).

**Tn-seq analysis**. Transposon mutant libraries of *A. baylyi* were generated by electroporation of cells with pDL1093, a vector that allows for Kan$^R$ mini-Tn10 mutagenesis[38,39], and plating on kanamycin plates. Approximately 100,000 Tn mutant colonies were scraped off of plates and pooled to generate the input transposon mutant library. Subsequent sequencing (as described below) revealed that this input library contained 62,799 unique Tn insertions. This mutant library was then subjected to natural transformation in three separate replicates exactly as described above using three different sources of tDNA (1000 ng of each): a ΔACIAD1551::Spec$^R$ PCR product, an RpsL K43R (Sm$^R$) PCR product, or an RpoB PCR product from a spontaneous rifampicin-resistant mutant. Following the 5-h incubation with tDNA, a small portion of each transformation reaction was diluted and plated for quantitative culture on selective media (i.e., containing the appropriate antibiotic to select for transformants) to assess the number of transformants in each reaction. The remainder of the reactions were split and outgrown in selective medium overnight (i.e., with the appropriate antibiotic added to select for transformants = "transformant pool") or grown overnight in nonselective medium (i.e., in plain LB medium = "total pool"). There were >10⁷ total transformants obtained for each transformation reaction.

Sequencing libraries of the transposon-genomic junctions were generated for the Illumina platform using HTM-PCR exactly as previously described[39,40]. All sequencing data analysis was performed on the Galaxy platform[41] essentially as previously described[39]. First, reads were clipped to remove poly-C tails from the 3' ends of reads, which are generated during HTM-PCR. Next, reads were mapped to the *A. baylyi* ADP1 genome[42] using Bowtie with its default settings. Then, mapped reads were

further analyzed using Tufts TUCF genomics custom scripts that determine (1) the total number of unique reads that map to each gene (i.e., the number of unique Tn insertions in each gene in a given sample), and (2) the total number of reads that map to each gene (i.e., the relative abundance of Tn mutants in each gene in the overall cell population for a given sample). The total number of reads that map to each gene (i.e., the relative abundance of Tn insertions within a gene within the sample) is then normalized based on the relative size of each gene and the total number of mapped reads for the sample: ((number of reads that mapped to gene X / the total number of mapped reads for the sample)/(size of gene X / size of genome)). Thus, the expected normalized abundance for a "neutral" gene based on this standardization (i.e., a gene whose inactivation is not selected for or against) is 1.0.

Comparative analyses for Tn-seq were performed treating the "transformant pool" as the output and the "total pool" as the input to determine genes that were over- and under-represented following selection for transformants. Gene fitness was only assessed if a gene contained at least 1 transposon insertion in at least two out of the three replicates of the "total pool" samples. Also, the normalized abundance of insertions within a gene had to be greater than 0.01 in the "total pool" when averaged across all three replicates. These cutoffs allowed us to assess the phenotype of ~83% of the genes (2747/3310) in the A. baylyi ADP1 genome. For visualization, the relative abundance of Tn insertions within each gene is plotted from the "transformant pool" relative to the "total pool." See Supplementary Data 1 for the relative abundance of Tn insertions in each gene for all of the Tn-seq datasets analyzed for this manuscript. Also, the raw sequencing data for these Tn-seq experiments has been uploaded to the Sequence Read Archive (https://www.ncbi.nlm.nih.gov/sra): PRJNA716813.

**Pilin labeling, imaging, and quantification**. Pilin labeling was performed as described previously with some changes[5,14]. Briefly, 100 μL of overnight-grown cultures was added to 900 μL of LB in 1.5 mL microcentrifuge tube, and cells were grown at 30 °C rotating on a roller drum for 70 min. Cells were then centrifuged at 18,000 x g for 1 min and then resuspended in 50 μL of LB before labeling with 25 μg/mL of AlexaFluor488 $C_5$-maleimide (AF488-mal) (ThermoFisher) for 15 min at room temperature. Labeled cells were centrifuged, washed once with 100 μL of LB without disrupting the pellet, and resuspended in 5–20 μL LB. Cell bodies were imaged using phase-contrast microscopy while labeled pili were imaged using fluorescence microscopy on a Nikon TiE microscope using a Plan Apo 100X objective, a GFP filter cube for pili, a Hamamatsu ORCAFlash4.0 camera, and Nikon NIS Elements Imaging Software. Cell numbers and the percent of cells making pili were quantified manually using ImageJ[43]. All imaging was performed under 1% UltraPure agarose (Invitrogen) pads made with phosphate-buffered saline (PBS) solution. To image pili at different growth phases, 50 μL of overnight culture was labeled directly without dilution as the "overnight" growth point. To label the overnight "diluted" culture, 100 μL of overnight-grown cultures was added to 900 μL of LB, then immediately centrifuged and labeled as described above. To label pili in cells in the exponential growth phase, cells were grown for 70 min and labeled exactly as described above. To label mid to late exponentially growing cultures, cells were grown for 3 h before labeling.

**Western blotting**. Approximately $10^9$ cells from overnight cultures were concentrated into a pellet by centrifugation, and the culture supernatant was removed. Cell pellets were resuspended in 50 μL PBS and then mixed with an equal volume of 2× SDS-PAGE sample buffer (125 mM Tris, pH 6.8, 20% glycerol, 4% SDS, 0.4% bromophenol blue, and 10% β-mercaptoethanol) and boiled using a heat block set to 100 °C for 10–15 min. Proteins were separated on a 4–20% pre-cast polyacrylamide gel (Biorad) by SDS electrophoresis, electrophoretically transferred to a nitrocellulose membrane, and probed with 1:5000 dilution of mouse monoclonal α-FLAG antibodies (Sigma), 1:2500 dilution of mouse monoclonal α-GFP, and/or 1:12,000 dilution of mouse monoclonal α-RpoA (BioLegend) primary antibodies. Blots were then incubated with 1:10,000 dilution of α-mouse IRDye secondary antibodies (Licor) and imaged using a Licor imaging system.

**Coimmunoprecipitation (pulldown) experiments**. Overnight cultures of cells grown in tubes in LB medium were diluted by 1/10 into fresh LB for a total volume of 30 or 50 mL in 125- or 250-mL volume flasks, respectively. Then, 30–50 mL cultures were grown to exponential growth phase by shaking for 1.5 h at 30 °C. The total culture volume was then harvested at 10,000 x g for 10 min at room temperature, and the supernatant was removed. Cell pellets were resuspended in 2 mL of Buffer 1 (50 mM Tris-Cl pH 7.4, 150 mM NaCl, 1 mM EDTA) and transferred to 2 mL volume microcentrifuge tubes and centrifuged at 18,000 x g for 1 min. Cells were washed once more with 2 mL of Buffer 1, and washed pellets were resuspended in 1 mL of Buffer 2 (50 mM Tris-Cl pH 7.4, 150 mM NaCl, 1 mM EDTA, 10 mM $MgCl_2$, 0.1% Triton X-100, 2% glycerol). To lyse cells, 4200 units of Ready-Lyse lysozyme (Lucigen), 30 units of DNase I (New England Biolabs), and 10 μL of concentrated protease inhibitor cocktail (Sigma) (one pellet dissolved in 500 μL of Buffer 1) were added to cell suspensions and incubated at room temperature for 45 min. Cell debris was removed by centrifugation at 10,000 x g for 5 min at 4 °C. Then, 50 μL of cell lysates were set aside and used as an "input" sample; 50 μL aliquots of α-FLAG magnetic bead slurry (Sigma) or α-GFP magnetic bead slurry (MBL biotech) in 1.5 ml microcentrifuge tubes were washed three times with 1 mL

of Buffer 2 using a magnetic collection stand; and 1 mL of cell lysates was added to washed magnetic beads and subjected to end-over-end rotation at 4 °C for 2 h. Beads were then washed three times with 0.5 mL of Buffer 2, with 10 min incubations in Buffer 2 at 4 °C between each wash step. Beads were briefly washed a fourth time with 0.5 mL Buffer 2. To elute proteins from α-FLAG beads, 100 μL of elution buffer (150 μg/mL 3X-FLAG peptide, Sigma, in Buffer 2) was added and samples were subjected to end-over-end rotation at 4 °C for 30 min. To elute proteins from α-GFP beads, beads were resuspended in 50 μL PBS, 2× SDS-PAGE sample buffer was added to tubes, and samples were boiled as described above. Eluates and input samples were subjected to western blotting as described above.

**Measuring GFP fluorescence in reporter strains**. V. cholerae reporter strains harboring $P_{cpiA}$-GFP were grown to late log in LB medium with or without 0.2% arabinose as indicated. Cells were then washed once in instant ocean medium (7 g/L; Aquarium Systems) and transferred to a 96-well plate. Fluorescence was then determined on a Biotek H1M plate reader with monochromater set to 500 nm for excitation and 540 nm for emission.

**Fluorescent DNA binding/uptake**. A ~7 kb PCR product was fluorescently labeled as described previously[10] using the Cy3 LabelIT kit (Mirus Biosciences) as per the manufacturer's recommendations. For the parent strain, 1 μL (100 ng) of Cy3-DNA was added to 900 μL of LB in a 1.5-mL microcentrifuge tube along with 100 μL of overnight-grown cultures, and cells were grown at 30 °C rotating on a roller drum for 70 min. Pili were then labeled as described above.

To visualize DNA binding in pilT mutants, cells were grown as described above, but 1 μL (100 ng) of Cy3-DNA was added to cells along with AF488-mal and incubated for 15–25 min before washing. Cells were imaged using the same microscopy setup described above, using a dsRed filter cube to image Cy3-DNA.

**Phylogenetic analysis**. TfpB homologues were identified by BLAST using 43 manually selected Gammaproteobacteria genomes[44] through Integrated Microbial Genomes and Microbiomes online resources (IMG)[45] resulting in 209 protein sequences. The FtsK protein from A. baylyi ADP1 was added manually and used as an outgroup. Through the NGPhylogeny.fr server[46], sequences were aligned using the default parameters of MAFFT[47], and phylogenetic tree construction was performed on the aligned sequences using the default parameters of FastTree software[48,49] set to perform 100 bootstraps[50]. The resulting tree was visualized using the Interactive Tree of Life visualization software[51], from which trees were exported for publication.

**Statistics and reproducibility**. All experiments were repeated a minimum of two times, and all attempts at replication were successful. Data in Figs. 1a, 3d, and 4d, and Supplementary Figs. 4 and 8, are representative of two independent experiments. Data in Figs. 1b, 2c, and 4a are representative of three independent experiments. Data in Supplementary Fig. 1c is representative of dozens of independent experiments.

**Reporting summary**. Further information on research design is available in the Nature Research Reporting Summary linked to this article.

## Data availability
Tn-seq data are available on Sequence Read Archive (https://www.ncbi.nlm.nih.gov/sra) under accession code PRJNA716813. Source Data are provided with this paper.

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

## Acknowledgements

We would like to thank E. Geisinger for *A. baylyi* ADP1 ATCC33305 wildtype strain, and X. Charpentier for the apramycin resistance cassette. We would like to thank K. Hummels and B. Bratton for helpful suggestions on phylogenetic analysis. C. K. E. is a Damon Runyon Fellow supported by the Damon Runyon Cancer Research Foundation (DRG-2385-20). This work was supported in part by the National Science Foundation, through the Center for the Physics of Biological Function (PHY-1734030). This work was supported by grant R35GM128674 from the National Institutes of Health awarded to A. B. D. and the National Institutes of Health Pioneer Award 1DP1AI124669-01 awarded to Z. G.

## Author contributions

C.K.E. and A.B.D. designed and coordinated the overall study. C.K.E., T.N.D., C.A.K., and A.B.D. performed the experiments. All authors analyzed and interpreted data. C.K.E. and A.B.D. wrote the manuscript with help from Z.G. and J.W.S.

## Competing interests

The authors declare no competing interests.
