## [Peer Review File · Nature Communications]

REVIEWER COMMENTS

Reviewer #1 (Remarks to the Author):

This manuscript describes development of the highly transformable species *Acinetobacter baylyi* as a model for the investigation of type IV pilus-mediated competence. The authors showed using fluorescent labeling of Cys modified pilins that the bacteria produce multiple short pili along the long axis of the ovoid cells that co-localize with labeled eDNA. A Tn-seq experiment to identify genes involved in transformation uncovered many known genes but also a second PilB-like ATPase and a pair of genes encoding a putative regulator and putative inhibitor, CpiR-CpiA. A series of epistasis studies showed that TfpB is the main ATPase responsible for pilus assembly, while PilB plays a secondary role. The authors show that CpiR regulates expression of CpiA, which is proposed to inhibit PilB function through direct interaction, supported by pulldown data. Overall, this is a well written, well presented, and interesting study and I have only minor suggestions for improvement. In particular, the discussion could have been more fulsome.

Comments.

Line 46 'much shorter' is not informative – it would be helpful to readers less familiar with the pilus field to provide a typical range of pilus lengths and compare them to those seen for this species. What is the hypothesis regarding the localization of pili to non-polar positions in this species (similar to cyanobacteria that use pili for phototaxis, perhaps a discussion point)?

Line 71 and Fig 2 – use quantitative terms when expressing the change in transformation efficiency in the pilB mutant. How do the authors reconcile the difference in piliation between the pilB and tfpB mutants with the similarity in transformation efficiency in the two backgrounds? If both motors are required for full efficiency, how do the authors envision both engaging with the machinery? Does *A. baylyi* express a type II secretion system?

Supplemental Fig 1 – what is PilB2? Is this TfpB or another ATPase?

Supplemental Fig 2 – use a symbol to highlight TfpB. Explain in the legend that 'PilQ' in this context is not the secretin component to avoid confusion.

Supplemental Fig 3 – the difference in transformation frequency between a pilB deletion and a cpiR deletion (which should lead to inhibition of pilB) seem reversed; it looks like a pilB delete background has a higher frequency, whereas I might have predicted that it would be a more thorough form of inhibition than expression of CpiA. It's interesting that CpiA is repressed – how does this set up compare with what is seen with phage encoded inhibitors of PilB? Are there any similarities in predicted structure despite the lack of sequence identity?

Supplemental Fig 4 – the title of this figure should be reworded. The data support the interaction but there is only correlative evidence that PilB activity is inhibited. Its interaction with other components could be blocked rather than its activity as an ATPase.

Reviewer #2 (Remarks to the Author):

Ellison et al present a study characterizing components affecting transformability of *Acinetobacter baylyi*. Using transposon mutagenesis, they identify genes involved in transformation. While most of these genes have been shown to affect transformability in other species, they find two new components. First, they show that next of PilB, a second ATPase drives type 4 pilus elongation. They name this protein TfpB. Second, they find a regulator, CpiR, and show that it inhibits pilus formation, most likely by repressing pilB.

The work is interesting, timely, well conducted, and shows a very important result: there is more than one type 4 pilus elongation motor. However, in my opinion, there is little conceptual advance on understanding of bacterial transformation. Therefore, I recommend submitting this manuscript to a journal specialized in microbiology. Moreover, a recent publication characterizing

transformation in a closely related species, *Acinetobacter baumannii*, has substantial overlap with the present study. (see Vesel & Blokesch, J. Bacteriol., 2021)

Major points:

1. Please refer to the recent publication by Vesel & Blokesch, J. Bacteriol., 2021 about transformation of *A. baumannii* and compare your findings.

Minor points:

2. The pili shown in this manuscript are very short. It might be interesting to investigate whether piliation is growth phase dependent.

3. line 127 Supplemental Figure 3 -> 4

4. In Fig. 2a the authors show that the transformation frequency is strongly reduced in the Δ pilB strain but Fig. 2b shows that the fraction of piliated cells is hardly affected. How do the authors explain this discrepancy?

Reviewer #3 (Remarks to the Author):

In this paper, the authors used a Tnseq approach to identify genes which products are implicated in the transformability of *Acinetobacter baylii*, and study the function of 2 of these genes TfpB and CpiA. They show some interesting results, but they are not as strong or as clear-cut as it is stated by the authors. More experiments are needed to confirm certain conclusions.

Major comment / Tnseq:

The Tnseq analysis which is the base of their results is not properly described (l200-221). The authors speak of 100 000 obtained mutants: do they mean unique mutants? Is the library saturated? The authors use different type of transforming DNA to screen the library: what is the transformation efficiency of the WT strain for each of these experiment (a low efficiency would create a bottleneck effect that would impact the results of the selection)? The authors say that they mapped the data on *V.cholerae* genome: is that a simple typo (it should be *A.baylii*)? The authors don't explain the tools used to determine the relative abundance of each insertion ("the Galaxie platform", l213, can be used to implement multiple tools, which ones did they used?). The author don't provide a proper statistical analysis of the results: the explanations given l215-219 are not clear and incomplete. The only result presented by the authors is a graphical representation: a table of results containing at least the statistically significant over- and under-represented genes (names, role and ACIAD numbers) should be included.

Major comment / TfpB-PilB:

Looking at the transformation assays, it seems that the T4P of *A.baylii* can use 2 non-phylogenetically-related PilB homologs. The 2 proteins seem to have a somewhat redundant role and are not "both essential" as stated by the authors. I would say that the presence of both are necessary for optimal T4P extension. If we look at the transformation results: removing one protein does not have a strong effect on the transformation phenotype, but removing both is indeed drastic. However, the authors seem to rely more on the results of their microscopy experiments (l76-81, fig 2): "T4P labeling in pilB, tfpB, and pilB tfpB mutants revealed that all three mutants were defective in T4P synthesis while pilT was intact (Figure 2c). In a pilT deletion background, pilB mutant populations had similar numbers of cells producing T4P compared to the Δ pilT parent while, surprisingly, tfpB mutants were highly defective in T4P synthesis (Figure 2b, c). The pilB tfpB pilT triple mutant produced no detectable T4P fibers (Figure 2b, c). Together, these results indicate that both PilB and TfpB are essential for efficient T4P extension in *A. baylyi*, with TfpB playing the dominant role." If indeed the "three mutants were defective in T4P synthesis" then there would be no possible transformation in these mutants, but that's not the case! Not seeing the pili does not mean they are not there, maybe they are shorter or less stable during imaging or something else. Even when we look at the WT pilus (fig 1a) the AF488-mal label is not super convincing by itself to show that the small foci are pili -> it is convincing only when comparing with the localization of the fluorescent DNA. The authors should show images of Δ pilB / Δ tfpB / Δ pilB, Δ tfpB cells with fluorescent DNA. Also, in general, microscopy images should show

more than one chosen cell, they can be put as supplemental but they should be somewhere. The authors did not try to complement each mutant with a surexpression of the other protein so what they observe could simply be due to a dose effect. For example, if we say that in WT, there are 100 PilB and 100 TfpB produced, and these 200 proteins allow for efficient T4P. In a Δ pilB or a Δ tfpB, there are only 100 proteins and that's not enough, hence the partial loss of transformability. The authors should consider studying the transformability of a " Δ pilB, Ptac-tfpB" and a " Δ tfpB, Ptac-pilB" strains. If indeed PilB and TfpB are necessary (because they are not perfectly redundant), then one should not complement the other.

The authors rely on a partial phylogenetic analysis (only "43 manually selected Gammaproteobacteria") of PilB homologs to state that (1) "TfpB is highly conserved in other Acinetobacter species" -> Where are the results showing this "high conservation"? An analysis in all Acinetobacter genomes should be done to show that. And (2) "proteins that cluster with TfpB may play similar functions in other bacterial species and that multiple extension ATPases may be a common feature of diverse T4P"-> Did they analyse the genomes containing a TfpB protein to see whether or not a PilB protein was also present? Maybe some species have a T4P where TfpB is alone? It seems to me that most species don't have TfpB and still have perfectly functional T4P. So, I agree that the presence of 2 extension motors for a T4P is original but it seems to be, for now at least, a property of only Acinetobacter (if it is proven that TfpB is indeed highly conserved in Acinetobacter).

Major comment / CpiA:

The authors show convincingly that CpiA can indeed inhibit PilB function and that they interact together. They don't hypothesized beyond that. What could the mechanism of this inhibition? Is CpiA delocalizing PilB, is it blocking the ATP site, is it competing with other PilB partners, is it modifying PilB, is there other exemple of such inhibitor, is it specific to Acinetobacter? It is not necessary for the authors to know all this but it would be nice to have a bit more possibilities discussed in the paper.

Minor comments:

- l13 & l17: the abstract should not contain references
- sup fig 1a: The drawing is nice but because there's so many genes, it is sometimes not clear what is represented where. A table with the relevant genes, their ACIAD number and their function would be nice.
- l36: ref?
- l47: long T4P filament have also been shown in *L.pneumophila* (Hardy et al 2021 (doi: 10.1128/JB.00548-20))
- l52: the Tnseq approach was used multiple times to identify genes involved in transformation, cited them would be relevant: Dalia et al 2014 (doi: 10.1128/mBio.01028-13), Taton et al 2020 (doi: 10.1038/s41467-020-15384-9), Hardy et al 2021 (doi: 10.1128/JB.00548-20)
- fig 1: the labels a, b, c, d should be rearranged for clarity -> d should be b, b should be c, c should be d
- fig 1b: the zoom-in region is not drawn properly in the fullsize graph (for exemple, pilC is outside the square box in the graph but is shown in the zoom in)
- fig 1d: it should be noted on the image and in the legend that the "parent" strain is Δ pilT
- l71: a figure of the different genomic arrangement would be nice
- fig 2a: it would be helpful to state in the legend that ComP is a major pilin so the mutant is used as a negative control
- l98: "Transcriptional repressors are often transcribed immediately adjacent to the genes they repress" is there a reference for that statement?
- l102-103: "further suggesting that CpiR represses cpiA transcription." cite fig 3b
- l169-170: is it 3kb like upstream?
- l212: it should be *A. baylyi* ADP1 not *V. cholerae*

We thank both reviewers and the editor for carefully evaluating our initial submission and providing constructive feedback. We have addressed all reviewer comments and concerns in the revised manuscript by making the suggested changes to the text and Figures. Also, we have performed additional experiments based on reviewer feedback that provide additional supporting evidence for the model presented. All major changes / additions to the main text are shown in red text in the revised manuscript. Please see below for a detailed response to each reviewer comment.

REVIEWER COMMENTS

Reviewer #1 (Remarks to the Author):

This manuscript describes development of the highly transformable species *Acinetobacter baylyi* as a model for the investigation of type IV pilus-mediated competence. The authors showed using fluorescent labeling of Cys modified pilins that the bacteria produce multiple short pili along the long axis of the ovoid cells that co-localize with labeled eDNA. A Tn-seq experiment to identify genes involved in transformation uncovered many known genes but also a second PilB-like ATPase and a pair of genes encoding a putative regulator and putative inhibitor, CpiR-CpiA. A series of epistasis studies showed that TfpB is the main ATPase responsible for pilus assembly, while PilB plays a secondary role. The authors show that CpiR regulates expression of CpiA, which is proposed to inhibit PilB function through direct interaction, supported by pulldown data. Overall, this is a well written, well presented, and interesting study and I have only minor suggestions for improvement. In particular, the discussion could have been more fulsome.

We thank the reviewer for their very positive and helpful comments for improving the clarity and discussion for our manuscript. We have incorporated the reviewer's suggestions and feel this has greatly improved the paper. See below for specifics:

Comments.

Line 46 'much shorter' is not informative – it would be helpful to readers less familiar with the pilus field to provide a typical range of pilus lengths and compare them to those seen for this species. What is the hypothesis regarding the localization of pili to non-polar positions in this species (similar to cyanobacteria that use pili for phototaxis, perhaps a discussion point)?

We have expanded this section to include these points. We changed “T4P in *A. baylyi* appear much shorter than those found in other species like *V. cholerae*, and they localize close together in a line along the long axis of the cell” to “T4P in *A. baylyi* are much shorter, often appearing as puncta rather than the 1 μm long filaments found in other species like *V. cholerae* or the 10 μm long filaments found in *Legionella pneumophila*. *A. baylyi* TFP also localize close together in a line along the long axis of the cell, although the functional consequences of this localization pattern remain unclear.” (lines 46-50)

Line 71 and Fig 2 – use quantitative terms when expressing the change in transformation efficiency in the pilB mutant. How do the authors reconcile the difference in piliation between the pilB and tfpB mutants with the similarity in transformation efficiency in the two backgrounds?

As suggested, we have added a line to be more quantitative about expressing changes in transformation in the *pilB* and *tfpB*: “Mutation of *tfpB* reduced transformation rates by two orders of magnitude, similar to the *pilB* mutant” (lines 80-81).

We have also performed additional experiments (new supplemental figures 4 and 5) that may explain the difference between piliation phenotypes and transformation frequency between these

two mutants. These data support the model that PilB and TfpB are playing functionally distinct roles in pilus extension, resulting in differences in piliation phenotypes in *pilT* mutants. However, our data also suggest these motors can play partially redundant roles to compensate for the loss of each other, which may explain their similar rates of transformation. We have added some discussion in addition to the new data to try to address this point: “T4P labeling in *pilB*, *tfpB*, and *pilB tfpB* mutants revealed that none of these three mutants produced visible pili while *pilT* was intact (Figure 2c). Because T4P are required for natural transformation, we hypothesized that *pilB* or *tfpB* mutants, which are still highly transformable, may either (1) make short pili that cannot be resolved by our fluorescence microscopy based approach or (2) exhibit reduced rates of pilus dynamic activity that result in a reduction in detectable surface pili in static images. To test the latter hypothesis, deletion of *pilT* prevents T4P retraction and can thus reveal more subtle effects on pilus extension since all T4P made by a cell remain in an assembled state. In a *pilT* deletion background, *pilB* mutant populations had similar numbers of cells producing T4P compared to the Δ *pilT* parent while, surprisingly, *tfpB* mutants were highly defective in T4P synthesis (Figure 2b, c). The *pilB tfpB pilT* triple mutant produced no detectable T4P fibers (Figure 2b, c). These data suggest that PilB and TfpB may play distinct roles in T4P extension. Because *tfpB pilT* mutants make very few T4P, but *pilB pilT* mutants make similar levels of surface pili compared to the Δ *pilT* parent, we hypothesize that TfpB plays a larger role in initiating T4P extension, while PilB plays a larger role in driving processive T4P extension. In this model, each motor must still be able to partially compensate for the loss of the other to achieve some degree of T4P extension, which would account for the observed similarities in natural transformation frequencies when comparing *pilB* and *tfpB* single mutants.” (lines 83-98).

If both motors are required for full efficiency, how do the authors envision both engaging with the machinery? Does *A. baylyi* express a type II secretion system?

We have added text to the discussion to address this point: “The discovery that two extension motors drive T4P synthesis in *A. baylyi* raises an interesting biological question. How might two extension motors engage with the machinery during extension? Previous work has shown that multiple retraction motors can work together to drive forceful T4P retraction and while the nature of how these motors are able to simultaneously interact with the machinery remains unknown, a similar mechanism may exist for PilB and TfpB. Alternatively, each motor may play temporally distinct roles during extension that allow each motor complex to interact with the machinery separately. Our data show that *tfpB* mutants make substantially fewer pili than *pilB* mutants in the absence of retraction, suggesting that TfpB may play a role in initiating T4P extension while PilB may promote processive pilus extension. In this model, each motor possesses a distinct function that could be accomplished by temporally segregated machinery interactions.” (lines 183-192)

A. baylyi does possess a type II secretion system, but we have found that deletion of type II secretion components does not affect type IV pilus synthesis. This is also supported our Tn-seq, which indicates that T2SS machinery components (Xcp genes) are not defective at natural transformation (see newly included Spreadsheet 1).

Supplemental Fig 1 – what is PilB2? Is this TfpB or another ATPase?

We thank the reviewer for catching this typo. PilB2 has been changed to TfpB.

Supplemental Fig 2 – use a symbol to highlight TfpB. Explain in the legend that ‘PilQ’ in this context is not the secretin component to avoid confusion.

We have added a symbol to highlight TfpB, and we have fixed the legend for clarity that PilQ here is a motor: “PilQ in this context is not the secretin component, but is an alternative name for motors found in some species.”

Supplemental Fig 3 – the difference in transformation frequency between a pilB deletion and a cpiR deletion (which should lead to inhibition of pilB) seem reversed; it looks like a pilB delete background has a higher frequency, whereas I might have predicted that it would be a more thorough form of inhibition than expression of CpiA.

These data are not statistically significantly different ($P = 0.7808$ by Tukey’s multiple comparisons test) suggesting that both a *pilB* or *cpiR* deletion affect transformation to the same extent. These results are consistent with CpiA-dependent inhibition of PilB activity as proposed in the manuscript.

It’s interesting that CpiA is repressed – how does this set up compare with what is seen with phage encoded inhibitors of PilB? Are there any similarities in predicted structure despite the lack of sequence identity?

This is an excellent question. To address this, we have added to the discussion: “CpiA is poorly modeled by structural prediction algorithms like Phyre2. But it is predicted that CpiA has a secondary structure rich in alpha helices, similar to a recently described phage-encoded PilB inhibitor, Aqs1. Although Aqs1 and CpiA are dissimilar at the sequence level, it is possible they share structural similarities that enable PilB binding and inhibition. In *Acinetobacter baylyi*, *cpiA* is encoded on the chromosome, and it does not appear to be found in other sequenced species of *Acinetobacter*. Thus, it is tempting to speculate that CpiA was acquired after phage infection of an ancestral strain and co-opted for regulation of T4P activity. While expression of phage-encoded PilB-inhibitors like Aqs1 are not repressed in the same manner as CpiA, it could be that the CpiR repressor was acquired after CpiA to regulate its effects on T4P extension.” (lines 201-210)

Supplemental Fig 4 – the title of this figure should be reworded. The data support the interaction but there is only correlative evidence that PilB activity is inhibited. Its interaction with other components could be blocked rather than its activity as an ATPase.

We have fixed the title of this figure. We changed “CpiA specifically interacts with PilB to inhibit its activity” to “CpiA specifically interacts with PilB.”

Reviewer #2 (Remarks to the Author):

Ellison et al present a study characterizing components affecting transformability of *Acinetobacter baylyi*. Using transposon mutagenesis, they identify genes involved in transformation. While most of these genes have been shown to affect transformability in other species, they find two new components. First, they show that next of PilB, a second ATPase drives type 4 pilus elongation. They name this protein TfpB. Second, they find a regulator, CpiR, and show that it inhibits pilus formation, most likely by repressing pilB.

The work is interesting, timely, well conducted, and shows a very important result: there is more than one type 4 pilus elongation motor. However, in my opinion, there is little conceptual advance on understanding of bacterial transformation. Therefore, I recommend submitting this manuscript to a journal specialized in microbiology. Moreover, a recent publication characterizing transformation in a closely

related species, *Acinetobacter baumannii*, has substantial overlap with the present study. (see Vesel & Blokesch, J. Bacteriol., 2021)

We thank the reviewer for their positive comments about the importance of our work. However, we disagree that there is little conceptual advance on understanding bacterial transformation. As type IV pili are a critical component of DNA uptake preceding natural transformation, we feel these results are significant and will be of interest to those studying this process. Additionally, as T4P are found in several bacterial species including human pathogens, the discovery of a new class of extension motors and a motor inhibitor will be of broad interest to those studying related nanomachines and the diverse processes they mediate including virulence, biofilm formation, twitching motility, and surface sensing.

We also disagree that there is substantial overlap with the recent paper by Vesel and Blokesch. This paper was published only days before our initial submission, and while it nicely demonstrates growth phase-dependent production of type IV pili in *A. baumannii*, this paper does not explore the use of multiple motors or motor inhibitors in type IV pilus production or function.

Major points:

1. Please refer to the recent publication by Vesel & Blokesch, J. Bacteriol., 2021 about transformation of *A. baumannii* and compare your findings.

Minor points:

2. The pili shown in this manuscript are very short. It might be interesting to investigate whether piliation is growth phase dependent.

We have added data (Supplemental figure 2) to address this point, along with additional discussion to put these results into context with the study by Vesel and Blokesch, which is now cited in the revised manuscript as suggested. We show that pili are constitutively made in line with previous data showing that *A. baylyi* are transformable at different growth phases. We added: “Recent work has shown that T4P synthesis in *A. baumannii* is growth-phase dependent, but we find here that the T4P of *A. baylyi* are constitutively made (Supplemental figure 2), in line with previous findings that *A. baylyi* cells are transformable throughout all growth phases.” (lines 50-53)

3. line 127 Supplemental Figure 3 -> 4

We have added a reference to this supplemental figure.

4. In Fig. 2a the authors show that the transformation frequency is strongly reduced in the DpilB strain but Fig. 2b shows that the fraction of piliated cells is hardly affected. How do the authors explain this discrepancy?

We have added a few sentences to try to clarify the discrepancy between piliation in *pilT* mutants and effects on natural transformation. We hypothesize that PilB and TfpB are playing distinct roles in T4P extension but that they can partially compensate for deletion of the other, and we have added new data (supplemental figures 4 and 5) to support this hypothesis along with the following text: “T4P labeling in *pilB*, *tfpB*, and *pilB tfpB* mutants revealed that none of these three mutants produced visible pili while *pilT* was intact (Figure 2c). Because T4P are required for natural transformation, we hypothesized that *pilB* or *tfpB* mutants, which are still highly transformable, may either (1) make short pili that cannot be resolved by our fluorescence microscopy based approach or (2) exhibit reduced rates of pilus dynamic activity that result in a reduction in detectable surface pili in static images. To test the latter hypothesis, deletion of *pilT* prevents T4P retraction and can thus reveal more subtle effects on pilus extension since all T4P made by a cell

remain in an assembled state. In a *pilT* deletion background, *pilB* mutant populations had similar numbers of cells producing T4P compared to the Δ *pilT* parent while, surprisingly, *tfpB* mutants were highly defective in T4P synthesis (Figure 2b, c). The *pilB tfpB pilT* triple mutant produced no detectable T4P fibers (Figure 2b, c). These data suggest that PilB and TfpB may play distinct roles in T4P extension. Because *tfpB pilT* mutants make very few T4P, but *pilB pilT* mutants make similar levels of surface pili compared to the Δ *pilT* parent, we hypothesize that TfpB plays a larger role in initiating T4P extension, while PilB plays a larger role in driving processive T4P extension. In this model, each motor must still be able to partially compensate for the loss of the other to achieve some degree of T4P extension, which would account for the observed similarities in natural transformation frequencies when comparing *pilB* and *tfpB* single mutants. While T4P labeling in *pilT* mutant backgrounds supports this model (Figure 2b, c), it does not rule out the possibility that *tfpB pilT* mutants produce many short T4P that cannot be resolved by our pilus labeling approach. To test this, we incubated extension motor mutants with fluorescently labeled DNA, reasoning that DNA binding would reveal short T4P synthesized in *tfpB pilT* mutants. Fluorescent DNA did not bind the *tfpB pilT* mutant above the background level observed in the *pilB tfpB pilT* mutant except where T4P were visible (Supplemental figure 4), suggesting that short pili are not produced in the *tfpB pilT* strain, which further supports the model that TfpB plays a distinct role from PilB in initiating T4P extension.

Alternatively, it is possible that these motors are completely redundant, and the difference in phenotypes observed between *pilB* and *tfpB* mutants is simply due to a dose-dependent loss of motor protein expression. To test this, we sought to determine whether overexpression of *tfpB* was sufficient to compensate for a *pilB* deletion or whether overexpression of *pilB* was sufficient to compensate for a *tfpB* deletion. If these motors are completely redundant, we expected that overexpression of a motor would compensate for loss of the other motor and result in parental levels of natural transformation; however if these motors play distinct roles in T4P extension, overexpression would not be able to cross compensate and transformation frequencies would remain unchanged. When we performed this experiment, we found that overexpression of *pilB* did not restore the transformation defect of a *tfpB* mutant, and correspondingly, overexpression of *tfpB* did not restore the transformation defect of a *pilB* mutant (Supplemental figure 5). Together, these results indicate that both PilB and TfpB are essential for optimal T4P production in *A. baylyi*, with each motor playing a distinct role in T4P extension.” (lines 83-117).

Reviewer #3 (Remarks to the Author):

In this paper, the authors used a Tnseq approach to identify genes which products are implicated in the transformability of *Acinetobacter baylyi*, and study the function of 2 of these genes TfpB and CpiA. They show some interesting results, but they are not as strong or as clear-cut as it is stated by the authors. More experiments are needed to confirm certain conclusions.

We thank the reviewer for their comments and feedback on our manuscript. We have addressed each comment, and we feel this has very nicely bolstered our conclusions and the clarity of our work. See below for specific responses to each point:

Major comment / Tnseq:

The Tnseq analysis which is the base of their results is not properly described (1200-221). The authors speak of 100 000 obtained mutants: do they mean unique mutants? Is the library saturated? The authors use different type of transforming DNA to screen the library: what is the transformation efficiency of the WT strain for each of these experiment (a low efficiency would create a bottleneck effect that would

impact the results of the selection)? The authors say that they mapped the data on *V.cholerae* genome: is that a simple typo (it should be *A.baylii*)? The authors don't explain the tools used to determine the relative abundance of each insertion ("the Galaxie platform", l213, can be used to implement multiple tools, which ones did they used?). The author don't provide a proper statistical analysis of the results: the explanations given l215-219 are not clear and incomplete. The only result presented by the authors is a graphical representation: a table of results containing at least the statistically significant over- and under- represented genes (names, role and ACIAD numbers) should be included.

We thank the reviewer for their direction on how to make our Tnseq data clearer. We have substantially altered the Tnseq methods section to include more details on how libraries were generated, subjected to selection, and analyzed. We also include a supplemental spreadsheet as suggested that includes the relative abundance of Tn insertions for each gene in all of the experimental conditions tested. And finally, we have uploaded the raw sequencing reads to the Sequence Read Archive under accession number PRJNA716813. "Sequencing libraries of the transposon-genomic junctions were generated for the Illumina platform using HTM-PCR exactly as previously described. All sequencing data analysis was performed on the Galaxy platform⁴¹ essentially as previously described³⁹. First, reads were clipped to remove poly-C tails from the 3' ends of reads, which are generated during HTM-PCR. Next, reads were mapped to the *A. baylyi* ADP1 genome⁴² using Bowtie with its default settings. Then, mapped reads were further analyzed using Tufts TUCF genomics custom scripts that determine (1) the total number of unique reads that map to each gene (i.e. the number of unique Tn insertions in each gene in a given sample), and (2) the total number of reads that map to each gene (i.e. the relative abundance of Tn mutants in each gene in the overall cell population for a given sample). The total number of reads that map to each gene (i.e. the relative abundance of Tn insertions within a gene within the sample) is then normalized based on the relative size of each gene and the total number of mapped reads for the sample: ((number of reads that mapped to gene X / the total number of mapped reads for the sample)/(size of gene X / size of genome)). Thus, the expected normalized abundance for a "neutral" gene based on this standardization (i.e. a gene whose inactivation is not selected for or against) is 1.0.

Comparative analyses for Tn-seq were performed treating the "transformant pool" as the output and the "total pool" as the input to determine genes that were over- and under-represented following selection for transformants. Gene fitness was only assessed if a gene contained at least 1 transposon insertion in at least two out of the three replicates of the "total pool" samples. Also, the normalized abundance of insertions within a gene had to be greater than 0.01 in the "total pool" when averaged across all three replicates. These cutoffs allowed us to assess the phenotype of ~83% of the genes (2747/3310) in the *A. baylyi* ADP1 genome. For visualization, the relative abundance of Tn insertions within each gene is plotted from the "transformant pool" relative to the "total pool". See Spreadsheet 1 for the relative abundance of Tn insertions in each gene for all of the Tn-seq datasets analyzed for this manuscript. Also, the raw sequencing data for these Tn-seq experiments has been uploaded to the Sequence Read Archive (<https://www.ncbi.nlm.nih.gov/sra>): PRJNA716813." (lines 285-309)

Major comment / TfpB-PilB:

Looking at the transformation assays, it seems that the T4P of *A.baylii* can use 2 non-phylogenetically-related PilB homologs. The 2 proteins seem to have a somewhat redundant role and are not "both essential" as stated by the authors. I would say that the presence of both are necessary for optimal T4P extension. If we look at the transformation results: removing one protein does not have a strong effect on the transformation phenotype, but removing both is indeed drastic. However, the authors seem to rely more on the results of their microscopy experiments (176-81, fig 2): "T4P labeling in pilB, tfpB, and pilB tfpB mutants revealed that all three mutants were defective in T4P synthesis while pilT was intact (Figure

2c). In a *pilT* deletion background, *pilB* mutant populations had similar numbers of cells producing T4P compared to the Δ *pilT* parent while, surprisingly, *tfpB* mutants were highly defective in T4P synthesis (Figure 2b, c). The *pilB tfpB pilT* triple mutant produced no detectable T4P fibers (Figure 2b, c). Together, these results indicate that both PilB and TfpB are essential for efficient T4P extension in *A. baylyi*, with TfpB playing the dominant role.” If indeed the “three mutants were defective in T4P synthesis” then there would be no possible transformation in these mutants, but that’s not the case! Not seeing the pili does not mean they are not there, maybe they are shorter or less stable during imaging or something else. Even when we look at the WT pilus (fig 1a) the AF488-mal label is not super convincing by itself to show that the small foci are pili -> it is convincing only when comparing with the localization of the fluorescent DNA. The authors should show images of Δ *pilB* / Δ *tfpB* / Δ *pilB*, Δ *tfpB* cells with fluorescent DNA. Also, in general, microscopy images should show more than one chosen cell, they can be put as supplemental but they should be somewhere.

We have changed the wording of this section to make it clearer that we do not mean extension motor mutants are unable to make T4P. Specifically, we now state: “T4P labeling in *pilB*, *tfpB*, and *pilB tfpB* mutants revealed that none of these three mutants produced visible pili while *pilT* was intact (Figure 2c). Because T4P are required for natural transformation, we hypothesized that *pilB* or *tfpB* mutants, which are still highly transformable, may either (1) make short pili that cannot be resolved by our fluorescence microscopy based approach or (2) exhibit reduced rates of pilus dynamic activity that result in a reduction in detectable surface pili in static images.” (lines 83-88)

We appreciate the reviewer’s suggestion to add data for motor mutants incubated with DNA and feel this has bolstered our conclusions and strengthened the manuscript. We have added the suggested experiment and find that *tfpB* mutants do not bind fluorescently labeled DNA except where T4P are visible, supporting the idea that TfpB and PilB play distinct roles in T4P extension. We have included these data as larger fields of view as the new supplemental figure 4, and we have added text to expand on the interpretation of these data in the revised manuscript: “Because *tfpB pilT* mutants make very few T4P, but *pilB pilT* mutants make similar levels of surface pili compared to the Δ *pilT* parent, we hypothesize that TfpB plays a larger role in initiating T4P extension, while PilB plays a larger role in driving processive T4P extension. In this model, each motor must still be able to partially compensate for the loss of the other to achieve some degree of T4P extension, which would account for the observed similarities in natural transformation frequencies when comparing *pilB* and *tfpB* single mutants. While T4P labeling in *pilT* mutant backgrounds supports this model (Figure 2b, c), it does not rule out the possibility that *tfpB pilT* mutants produce many short T4P that cannot be resolved by our pilus labeling approach. To test this, we incubated extension motor mutants with fluorescently labeled DNA, reasoning that DNA binding would reveal short T4P synthesized in *tfpB pilT* mutants. Fluorescent DNA did not bind the *tfpB pilT* mutant above the background level observed in the *pilB tfpB pilT* mutant except where T4P were visible (Supplemental figure 4), suggesting that short pili are not produced in the *tfpB pilT* strain, which further supports the model that TfpB plays a distinct role from PilB in initiating T4P extension.” (lines 93-105)

The authors did not try to complement each mutant with a surexpression of the other protein so what they observe could simply be due to a dose effect. For example, if we say that in WT, there are 100 PilB and 100 TfpB produced, and these 200 proteins allow for efficient T4P. In a Δ *pilB* or a Δ *tfpB*, there are only 100 proteins and that’s not enough, hence the partial loss of transformability. The authors should consider studying the transformability of a “ Δ *pilB*, Ptac-*tfpB*” and a “ Δ *tfpB*, Ptac-*pilB*” strains. If indeed PilB and TfpB are necessary (because they are not perfectly redundant), then one should not complement the other.

We have performed the experiment suggested by the reviewer. These experiments showed that ectopic expression of each motor does not compensate for loss of the other motor (i.e.

overexpression of PilB cannot compensate for loss of TfpB, and vice versa). These data suggest that the phenotypes associated with each single mutant are not simply due to a dose-dependent loss of motor expression, but instead supports our hypothesis that TfpB and PilB play distinct roles in T4P extension. We have added these data as the new supplementary figure 5, along with text to discuss these findings. “Alternatively, it is possible that these motors are completely redundant, and the difference in phenotypes observed between *pilB* and *tfpB* mutants is simply due to a dose-dependent loss of motor protein expression. To test this, we sought to determine whether overexpression of *tfpB* was sufficient to compensate for a *pilB* deletion or whether overexpression of *pilB* was sufficient to compensate for a *tfpB* deletion. If these motors are completely redundant, we expected that overexpression of a motor would compensate for loss of the other motor and result in parental levels of natural transformation; however if these motors play distinct roles in T4P extension, overexpression would not be able to cross compensate and transformation frequencies would remain unchanged. When we performed this experiment, we found that overexpression of *pilB* did not restore the transformation defect of a *tfpB* mutant, and correspondingly, overexpression of *tfpB* did not restore the transformation defect of a *pilB* mutant (Supplemental figure 5). Together, these results indicate that both PilB and TfpB are essential for optimal T4P production in *A. baylyi*, with each motor playing a distinct role in T4P extension.” (lines 106-117)

The authors rely on a partial phylogenetic analysis (only “43 manually selected Gammaproteobacteria”) of PilB homologs to state that (1) “TfpB is highly conserved in other Acinetobacter species” -> Where are the results showing this “high conservation”? An analysis in all Acinetobacter genomes should be done to show that. And (2) “proteins that cluster with TfpB may play similar functions in other bacterial species and that multiple extension ATPases may be a common feature of diverse T4P”-> Did they analyse the genomes containing a TfpB protein to see whether or not a PilB protein was also present? Maybe some species have a T4P where TfpB is alone? It seems to me that most species don’t have TfpB and still have perfectly functional T4P. So, I agree that the presence of 2 extension motors for a T4P is original but it seems to be, for now at least, a property of only Acinetobacter (if it is proven that TfpB is indeed highly conserved in Acinetobacter).

We have softened our text about the high conservation of TfpB in Acinetobacter. We changed “TfpB is highly conserved in other Acinetobacter species” to “TfpB is conserved in the other Acinetobacter species analyzed here (Figure 2d, Supplemental figure 6)” (lines 124-125)

Additionally, we have added a table that includes the IMG Gene IDs of both *tfpB* and *pilB* for each species we analyzed where a TfpB homolog was identified. We added text to discuss this point, and highlight an example where TfpB is the only motor protein identified in the genome. “Interestingly, our phylogenetic analysis demonstrated that 15 of the 16 species that contained a TfpB motor homolog, also encoded a PilB motor homolog (Supplemental table 2). The exception to this observation, *Pseudomonas fluorescens* A506, possesses T4P machinery components, but lacks a *pilB* homolog and instead only harbors a *tfpB* homolog (Supplemental figure 3). This could suggest that TfpB is capable of acting as the sole extension motor in this species, though data are needed to test this hypothesis.” (lines 192-197)

Major comment / CpiA:

The authors show convincingly that CpiA can indeed inhibit PilB function and that they interact together. They don’t hypothesized beyond that. What could the mechanism of this inhibition? Is CpiA delocalizing PilB, is it blocking the ATP site, is it competing with other PilB partners, is it modifying PilB, is there other exemple of such inhibitor, is it specific to Acinetobacter? It is not necessary for the authors to know all this but it would be nice to have a bit more possibilities discussed in the paper.

We appreciate the reviewer's suggestions about possible mechanisms of how CpiA inhibits PilB and have included them in the manuscript. We added "The molecular mechanism of CpiA-dependent inhibition of PilB remains unclear. We demonstrate that CpiA binds to PilB. Thus, it is possible that (1) CpiA sequesters PilB to prevent it from interacting with machinery components, (2) that CpiA binds PilB to inhibit its ATPase activity, or (3) that CpiA modifies PilB to disrupt its function. Testing these hypotheses will be the focus of future work." (lines 211-214)

We also add a sentence about the specificity of *cpiA* for *A. baylyi*: "...and it does not appear to be found in other sequenced species of *Acinetobacter*" (line 206)

Minor comments:

- 113 & 117: the abstract should not contain references

We have removed references from the abstract.

- sup fig 1a: The drawing is nice but because there's so many genes, it is sometimes not clear what is represented where. A table with the relevant genes, their ACIAD number and their function would be nice.

We have added a table containing T4P biosynthesis genes, the protein name, and the predicted function (Supplemental table 1).

- 136: ref?

We have added a reference as suggested.

- 147: long T4P filament have also been shown in *L.pneumophila* (Hardy et al 2021 (doi: 10.1128/JB.00548-20)

We have included this reference. We changed "T4P in *A. baylyi* appear much shorter than those found in other species like *V. cholerae*, and they localize close together in a line along the long axis of the cell" to "T4P in *A. baylyi* are much shorter, often appearing as puncta rather than the 1 μm long filaments found in other species like *V. cholerae* or the 10 μm long filaments found in *Legionella pneumophila*" (lines 46-48)

- 152: the Tnseq approach was used multiple times to identify genes involved in transformation, cited them would be relevant: Dalia et al 2014 (doi: 10.1128/mBio.01028-13), Taton et al 2020 (doi: 10.1038/s41467-020-15384-9), Hardy et al 2021 (doi: 10.1128/JB.00548-20)

We have added these references as suggested.

- fig 1: the labels a, b, c, d should be rearranged for clarity -> d should be b, b should be c, c should be d

We have rearranged the labels as suggested.

- fig 1b: the zoom-in region is not drawn properly in the fullsize graph (for example, pilC is outside the square box in the graph but is shown in the zoom in)

We have corrected the box location as suggested.

- fig 1d: it should be noted on the image and in the legend that the "parent" strain is ΔpilT

These strains are all *pilT*-intact, and we have denoted this in the figure legend.

- 171: a figure of the different genomic arrangement would be nice

We have added a new figure (supplemental figure 3) to show gene synteny for *pilB* and *tfpB* loci for a few species from our phylogenetic analysis as suggested.

- fig 2a: it would be helpful to state in the legend that ComP is a major pilin so the mutant is used as a negative control

We have added this information to the legend as suggested.

- 198: “Transcriptional repressors are often transcribed immediately adjacent to the genes they repress“ is there a reference for that statement?

We have added references for this statement as suggested.

- 1102-103: “further suggesting that CpiR represses *cpiA* transcription.“ cite fig 3b

We have added the reference to this figure as suggested.

- 1169-170: is it 3kb like upstream?

This is correct – we have added this information to the methods.

- 1212: it should be *A. baylyi* ADP1 not *V. cholerae*

We thank the author for pointing out this typo and have fixed it.

REVIEWERS' COMMENTS

Reviewer #1 (Remarks to the Author):

This study describes the identification of a second PilB homolog, TfpB, in the *Acinetobacter baylyi* type IV pilus system as well as a regulator CpiR and inhibitor of PilB function, CpiA. The authors have carefully addressed the reviewers' comments and the manuscript is substantially improved. I have 2 minor points remaining.

Supplemental Fig 1 – switch PilN and PilO in the diagram, PilM and PilN interact, not PilM and PilO

Supplemental Fig 3 – to make this useful to readers, the genes need to be labeled with locus numbers and accession information. It may be useful to add a line in the discussion about the possible role of these tfpB homologs, since current evidence shows that some of the species in this figure use only PilB homologs for pilus extension.

Reviewer #2 (Remarks to the Author):

My specific comments have been addressed very well within the revised version and I have no further recommendations.

Reviewer #3 (Remarks to the Author):

I congratulate the authors on the amount and quality of additional experiments they provide for this revised version. In consequence, the claims are much better supported. In particular, the complementation experiments (sup fig 5), and the microscopy experiments with pilus and DNA labeling in the different mutants (sup fig 4) are especially crucial to better assess the non-redundant role of PilB and TfpB.

The text is also much clearer with additional information and a more interesting discussion.

I have only one minor comment:

L194-197: it would be nice to note here that *P. fluorescens* is indeed naturally transformable (with ref doi: 10.1128/AEM.67.6.2617-2621.2001). So it seems that a T4P with TfpB alone is enough, at least in this species.

REVIEWERS' COMMENTS

Reviewer #1 (Remarks to the Author):

This study describes the identification of a second PilB homolog, TfpB, in the *Acinetobacter baylii* type IV pilus system as well as a regulator CpiR and inhibitor of PilB function, CpiA. The authors have carefully addressed the reviewers' comments and the manuscript is substantially improved. I have 2 minor points remaining.

Supplemental Fig 1 – switch PilN and PilO in the diagram, PilM and PilN interact, not PilM and PilO

We have corrected the figure as suggested.

Supplemental Fig 3 – to make this useful to readers, the genes need to be labeled with locus numbers and accession information. It may be useful to add a line in the discussion about the possible role of these tfpB homologs, since current evidence shows that some of the species in this figure use only PilB homologs for pilus extension.

We have added a statement to the legend pointing the reader to the IMG accession numbers for both TfpB and PilB for each organism, which is provided in Supplementary table 2.

Reviewer #2 (Remarks to the Author):

My specific comments have been addressed very well within the revised version and I have no further recommendations.

Reviewer #3 (Remarks to the Author):

I congratulate the authors on the amount and quality of additional experiments they provide for this revised version. In consequence, the claims are much better supported. In particular, the complementation experiments (sup fig 5), and the microscopy experiments with pilus and DNA labeling in the different mutants (sup fig 4) are especially crucial to better assess the non-redundant role of PilB and TfpB.

The text is also much clearer with additional information and a more interesting discussion.

I have only one minor comment:

L194-197: it would be nice to note here that *P. fluorescens* is indeed naturally transformable (with ref doi: 10.1128/AEM.67.6.2617-2621.2001). So it seems that a T4P with TfpB alone is enough, at least in this species.

While some subspecies of *P. fluorescens* have been shown to be transformable, this has not been shown for the specific species we used in our bioinformatic analysis. Also, the strains in the paper cited by the reviewer do not appear to have publicly available genomes to

search for PilB or TfpB homologues. We therefore prefer to refrain from adding this point to the manuscript to err on the side of caution.